# Hierarchical Clustering:
# $O(1)$-Approximation for Well-Clustered Graphs*

**Bogdan-Adrian Manghiuc** ⓘD
School of Informatics
The University of Edinburgh
b.a.manghiuc@sms.ed.ac.uk

**He Sun**
School of Informatics
The University of Edinburgh
h.sun@ed.ac.uk

## Abstract

Hierarchical clustering studies a recursive partition of a data set into clusters of successively smaller size, and is a fundamental problem in data analysis. In this work we study the cost function for hierarchical clustering introduced by Dasgupta [12], and present two polynomial-time approximation algorithms: Our first result is an $O(1)$-approximation algorithm for graphs of high conductance. Our simple construction bypasses complicated recursive routines of finding sparse cuts known in the literature (e.g., [6, 11]). Our second and main result is an $O(1)$-approximation algorithm for a wide family of graphs that exhibit a well-defined structure of clusters. This result generalises the previous state-of-the-art [10], which holds only for graphs generated from stochastic models. The significance of our work is demonstrated by the empirical analysis on both synthetic and real-world data sets, on which our presented algorithm outperforms the previously proposed algorithm for graphs with a well-defined cluster structure [10].

## 1 Introduction

Hierarchical clustering (HC) studies a recursive partition of a data set into clusters of successively smaller size, via an effective binary tree representation. As a basic technique, hierarchical clustering has been employed as a standard package in data analysis, and has comprehensive applications in practice. While traditionally HC trees are constructed through bottom-up (agglomerative) heuristics, which lacked a clearly-defined objective function, Dasgupta [12] has recently introduced a simple objective function to measure the quality of a particular hierarchical clustering and his work has inspired a number of research on this topic [3, 6, 7, 8, 10, 11, 20, 24]. Consequently, there has been a significant interest in studying efficient HC algorithms that not only work in practice, but also have proven theoretical guarantees with respect to Dasgupta's cost function.

**Our contribution.** In this work, we present two new approximation algorithms for constructing HC trees that can be rigorously analysed with respect to Dasgupta's cost function. For our first result, we construct an HC tree of an input graph $G$ *entirely* based on the degree sequence of $V(G)$, and we show that the approximation guarantee of our constructed tree is with respect to the conductance of $G$, which will be defined formally in Section 2. The striking fact of this result is that, for any $n$-vertex graph $G$ with $m$ edges and conductance $\Omega(1)$ (a.k.a. expander graph), an $O(1)$-approximate HC tree of $G$ can be very easily constructed in $O(m + n \log n)$ time, although obtaining such result for general graphs is impossible under the Small Set Expansion Hypothesis (SSEH) [6]. Our theorem is in line with a sequence of results for problems that are naturally linked to the Unique Games and Small Set Expansion problems: it has been shown that such problems are much easier to solve once

---

*The full version of the paper is available at https://arxiv.org/abs/2112.09055.

35th Conference on Neural Information Processing Systems (NeurIPS 2021).

the input instance exhibits the high conductance property [4, 5, 16, 18]. However, to the best of our knowledge, our result is the first of this type for hierarchical clustering.

While our first result presents an interesting theoretical fact, we further study whether we can extend this $O(1)$-approximate construction to a much wider family of graphs occurring in practice. Specifically, we look at *well-clustered graphs*, i.e., the graphs in which vertices within each cluster are better connected than vertices between different clusters and the total number of clusters is constant. This includes a wide range of graphs occurring in practice with a clear cluster-structure, and have been extensively studied over the past two decades (e.g., [13, 15, 23, 26]). As our second and main result, we present a polynomial-time $O(1)$-approximation algorithm that constructs an HC tree for a well-clustered graph. Given that the class of well-clustered graphs includes graphs with clusters of different sizes and asymmetrical internal structure, our result significantly improves the previous state-of-the-art [10], which only holds for graphs generated from stochastic models. At the technical level, the design of our algorithm is based on the graph decomposition algorithm presented in [13], which is designed to find a good partition of a well clustered graph. However, our analysis suggests that, in order to obtain an $O(1)$-approximation algorithm, directly applying their decomposition isn't sufficient for our purpose. To overcome this bottleneck, we refine the output decomposition via a *pruning* technique, and carefully merge the refined parts to construct our final HC tree. In our point of view, our presented stronger graph decomposition procedure might have applications in other settings as well. To demonstrate the significance of our work, we compare our algorithm against the previous state-of-the-art with similar approximation guarantee [10] and well-known linkage heuristics on both synthetic and real-world data sets. Although our algorithm's performance is marginally better than [10] for the graphs generated from the stochastic block models (SBM), the cost of our algorithm's output is up to $50\%$ lower than the one from [10] when the clusters of the input graph have different sizes and some cliques are embedded into a cluster.

**Related work.**    Our work fits in a line of research initiated by Dasgupta [12], who introduced a cost function to measure the quality of an HC tree. Dasgupta proved that a recursive application of the algorithm for the Sparest Cut problem achieves an $O(\log^{3/2} n)$-approximation, which has been subsequently improved to $O(\log n)$ and $O(\sqrt{\log n})$ by [24] and [6, 11]. It is known to be NP-hard to find an optimal HC tree [12] , and SSEH-hard to achieve an $O(1)$-approximation for a general input instance [6] . Cohen-Addad et al. [10] studied a hierarchical extension of the SBM and showed that a certain SVD projection algorithm together with several linkage heuristics achieves a $(1 + o(1))$-approximation with high probability. We emphasise that our notion of well-clustered graphs generalises the SBM variant studied in [10] and does not assume the rigid hierarchical structure of the clusters.

For another line of related work, Moseley and Wang [20] studied the dual objective function and proved that `Average Linkage` achieves a $(1/3)$-approximation for the new objective function. Notice that, although this has received significant attention very recently [3, 7, 8, 9, 25], achieving an $O(1)$-approximation is tractable under this alternative objective. This suggests the fundamental difference on the hardness of the problem under different objective functions, and is our reason to entirely focus on the Dasgupta's cost function in this work.

## 2   Preliminaries

Throughout the paper, we always assume that $G = (V, E, w)$ is an undirected graph with $n$ vertices, $m$ edges and weight function $w : V \times V \to \mathbb{R}_{\geq 0}$. For any edge $e = \{u, v\} \in E$ we write $w_e$ or $w_{uv}$ to indicate the *similarity* weight between $u$ and $v$. For a vertex $u \in V$, we denote its degree by $d_u \triangleq \sum_{v \in V} w_{uv}$ and we assume that $w_{\max}/w_{\min} = O(\mathrm{poly}(n))$, where $w_{\min}(w_{\max})$ is the minimum (maximum) edge weight. We will use $d_{\min}, d_{\max}, d_{\mathrm{avg}}$ for the minimum, maximum and average degrees in $G$, where $d_{\mathrm{avg}} \triangleq \sum_{u \in V} d_u / n$. For a nonempty subset $S \subset V$, we define $G[S]$ to be the induced subgraph on $S$ and we denote by $G\{S\}$ the subgraph $G[S]$, where self loops are added to vertices $v \in S$ such that their degrees in $G$ and $G\{S\}$ are the same. For any two subsets $S, T \subset V$, we define the *cut value* $w(S, T) \triangleq \sum_{e \in E(S,T)} w_e$, where $E(S, T)$ is the set of edges between $S$ and $T$. For any set $S \subseteq V$, the *volume* of $S$ is $\mathrm{vol}(S) \triangleq \sum_{u \in S} d_u$ and we write $\mathrm{vol}(G)$ when referring

to $\mathrm{vol}(V(G))$. We further define the conductance of $S$ by

$$\Phi_G(S) \triangleq \frac{w(S, V \setminus S)}{\mathrm{vol}(S)},$$

and $\Phi_G \triangleq \min_{\substack{S \subset V \\ \mathrm{vol}(S) \leq \mathrm{vol}(G)/2}} \Phi_G(S)$. We call $G$ an expander graph if $\Phi_G = \Omega(1)$. For a graph $G$,
let $\mathbf{D} \in \mathbb{R}^{n \times n}$ be the diagonal matrix defined by $\mathbf{D}_{uu} = d_u$ for all $u \in V$. We denote by $\mathbf{A} \in \mathbb{R}^{n \times n}$
the adjacency matrix of $G$, where $\mathbf{A}_{uv} = w_{uv}$ for all $u, v \in V$. The *normalised Laplacian matrix* of
$G$ is defined as $\mathcal{L} \triangleq \mathbf{I} - \mathbf{D}^{-1/2}\mathbf{A}\mathbf{D}^{-1/2}$, where $\mathbf{I}$ is the identity $n \times n$ matrix. We will denote the
eigenvalues of $\mathcal{L}$ by $0 = \lambda_1 \leq \cdots \leq \lambda_n \leq 2$.

**Hierarchical clustering.** A *hierarchical clustering (HC) tree* of a given graph $G$ is a binary tree
$\mathcal{T}$ with $n$ leaf nodes such that each leaf corresponds to exactly one vertex $v \in V(G)$. Let $\mathcal{T}$ be
an HC tree of some graph $G = (V, E, w)$ and let $N \in \mathcal{T}$ be an arbitrary internal node[2] of $\mathcal{T}$. We
denote $\mathcal{T}[N]$ to be the subtree of $\mathcal{T}$ rooted at $N$, leaves $(\mathcal{T}[N])$ to be the set of leaf nodes of $\mathcal{T}[N]$
and $\mathrm{parent}_{\mathcal{T}}(N)$ to be the parent of node $N$ in $\mathcal{T}$. In addition, each internal node $N \in \mathcal{T}$ induces a
unique vertex set $C \subseteq V$ formed by the vertices corresponding to leaves $(\mathcal{T}[N])$. We will abuse the
notation and write $N \in \mathcal{T}$ for both the internal node of $\mathcal{T}$ and its corresponding set of vertices in $V$.

To measure the quality of an HC tree $\mathcal{T}$ with similarity weights, Dasgupta [12] introduced the
following cost function:

$$\mathrm{cost}_G(\mathcal{T}) \triangleq \sum_{e = \{u, v\} \in E} w_e \cdot |\mathrm{leaves}\left(\mathcal{T}[u \vee v]\right)|,$$

where $u \vee v$ is the lowest common ancestor of $u$ and $v$ in $\mathcal{T}$. Trees that achieve a better hierarchical
clustering have a lower cost, based on the following consideration: for any pair of vertices $u, v \in V$
that corresponds to an edge of high weight $w_{uv}$ (i.e., $u$ and $v$ are highly similar) a "good" HC tree
would separate $u$ and $v$ lower in the tree, thus reflected in a small size $|\mathrm{leaves}(\mathcal{T}[u \vee v])|$. We denote
by $\mathsf{OPT}_G$ the minimum cost of any HC tree of $G$, i.e., $\mathsf{OPT}_G = \min_{\mathcal{T}} \mathrm{cost}_G(\mathcal{T})$, and use $\mathcal{T}^*$ to
refer to an optimal tree achieving the minimum. We say that an HC tree $\mathcal{T}$ is an $\alpha$-approximate tree
if $\mathrm{cost}_G(\mathcal{T}) \leq \alpha \cdot \mathsf{OPT}_G$. All omitted proofs are deferred to the full version of this paper.

## 3   Hierarchical clustering for graphs of high conductance

In this section we study hierarchical clustering for graphs with high conductance and prove that, for
any input graph $G$ with $\Phi_G = \Omega(1)$, an $O(1)$-approximate HC tree of $G$ can be simply constructed
based on the degree sequence of $G$. As a starting point, we show that $\mathrm{cost}_G(\mathcal{T})$ for any $\mathcal{T}$ can be
upper bounded with respect to $\Phi_G$ and the degree distribution of $V(G)$.

**Lemma 3.1.** *It holds for any HC tree $\mathcal{T}$ of graph $G$ that* $\mathrm{cost}_G(\mathcal{T}) \leq \frac{9}{4\Phi_G} \cdot \min\left\{\frac{d_{\mathrm{avg}}}{d_{\mathrm{min}}}, \frac{d_{\mathrm{max}}}{d_{\mathrm{avg}}}\right\} \cdot$
$\mathsf{OPT}_G$.

While Lemma 3.1 holds for any graph $G$, it
implies some interesting facts for expander
graphs: first of all, when $G = (V, E, w)$ sat-
isfies $d_{\mathrm{max}}/d_{\mathrm{min}} = O(1)$ and $\Phi_G = \Omega(1)$,
Lemma 3.1 shows that any tree $\mathcal{T}$ is an $O(1)$-
approximate tree. In addition, although $\Phi_G$
plays a crucial role in analysing $\mathrm{cost}_G(\mathcal{T})$ as
for many other graph problems, Lemma 3.1 in-
dicates that the degree distribution of $G$ might
also have significant impact. One could natu-
rally ask the extend to which the degree distribu-
tion of $V(G)$ would influence the construction
of optimal trees.

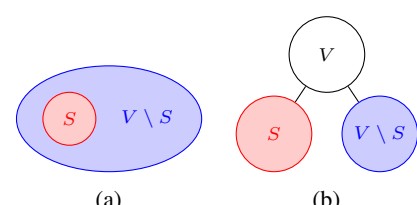

Figure 1: (a) Our considered graph $G$; (b) the tree that
separates the vertices of high degrees from the others
achieves $O(1)$-approximation.

To study this question, we look at the following graph $G$ where all edges have unit weight: (i) let
$G_1 = (V, E_1)$ be a constant-degree expander graph of $n$ vertices with $\Phi_{G_1} = \Omega(1)$, e.g., the ones

---
[2]We will always use *(internal) nodes* for the nodes of $\mathcal{T}$ and *vertices* for the elements of the vertex set $V$.

presented in [14]; (ii) we choose $\lfloor n^{2/3} \rfloor$ vertices from $V$ to form $S$, and let $K = (S, S \times S)$ be a complete graph defined on $S$; (iii) partition the vertices of $V \setminus S$ into $\lfloor n^{2/3} \rfloor$ groups of roughly the same size, associate each group to a unique vertex in $S$ and let $E_2$ be the set of edges formed by connecting every vertex in $S$ with all the vertices in its associated group. (iv) let $G \triangleq (V, E_1 \cup (S \times S) \cup E_2)$ be the union of $G_1$, $K$ and the edges in $E_2$, see Figure 1(a) for illustration. By construction, we know that $\Phi_G = \Omega(1)$, and the degrees of $G$ satisfy $d_{\max} = \Theta(n^{2/3})$, $d_{\min} = \Theta(1)$, and $d_{\mathrm{avg}} = \Theta(n^{1/3})$. Therefore, the ratio between $\mathrm{cost}_G(\mathcal{T})$ for any HC tree $\mathcal{T}$ and $\mathsf{OPT}_G$ could be as high as $\Theta(n^{1/3})$. On the other side, it's not difficult to show that the tree $\mathcal{T}^*$ illustrated in Figure 1(b), which first separates the set $S$ of high-degree vertices from $V \setminus S$ at the top of the tree, actually $O(1)$-approximates $\mathsf{OPT}_G$[3].

This example suggests that grouping vertices of similar degrees first could potentially help reduce $\mathrm{cost}_G(\mathcal{T})$ for our constructed $\mathcal{T}$. This motivates us to design the following Algorithm 1 to construct an HC tree. We highlight that the output of Algorithm 1 is uniquely determined by the ordering of the vertices of $G$ according to their degrees, which can be computed in $O(n \cdot \log n)$ time.

---

**Algorithm 1:** `HCwithDegrees`$(G\{V\})$

---

**Input:** $G = (V, E, w)$ with the ordered vertices such that $d_{v_1} \geq \ldots \geq d_{v_{|V|}}$;
**Output:** An HC tree $\mathcal{T}_{\mathrm{deg}}(G)$;
**1 if** $|V| = 1$ **then**
**2** $\quad$ **return** the single vertex in $V$ as the tree;
**3 else**
**4** $\quad$ $i_{\max} := \lfloor \log_2(|V| - 1) \rfloor$; $r := 2^{i_{\max}}$; $A := \{v_1, \ldots, v_r\}$; $B := V \setminus A$;
**5** $\quad$ Let $\mathcal{T}_1 = $ `HCwithDegrees`$(G\{A\})$; $\mathcal{T}_2 = $ `HCwithDegrees`$(G\{B\})$;
**6** $\quad$ **return** $\mathcal{T}_{\mathrm{deg}}$ with $\mathcal{T}_1$ and $\mathcal{T}_2$ as the two children.

---

**Theorem 1.** *Given any graph $G = (V, E, w)$ with conductance $\Phi_G$ as input, Algorithm 1 runs in $O(m + n \log n)$ time, and returns an HC tree $\mathcal{T}_{\mathrm{deg}}$ of $G$ that satisfies $\mathrm{cost}_G(\mathcal{T}_{\mathrm{deg}}) = O\left(1/\Phi_G^4\right) \cdot \mathsf{OPT}_G$.*

Theorem 1 guarantees that, when the input $G$ satisfies $\Phi_G = \Omega(1)$, the output $\mathcal{T}_{\mathrm{deg}}$ of Algorithm 1 achieves an $O(1)$-approximation. Moreover, as the high-conductance property can be determined in nearly-linear time by computing $\lambda_2(\mathcal{L}_G)$ and applying the Cheeger inequality, Algorithm 1 presents a very simple construction of an $O(1)$-approximate HC tree once the input $G$ is known to have high conductance.

**Proof sketch of Theorem 1.** The key notion employed in our proof is that of the *dense branch* which can be informally described as follows: we perform a traversal in $\mathcal{T}$ starting at the root node $A_0$ and we sequentially travel to the child of *higher* volume. The process stops whenever we reach a node $A_k$, for some $k \in \mathbb{Z}_{\geq 0}$, such that $\mathrm{vol}(A_k) > \mathrm{vol}(G)/2$ and both its children have volume at most $\mathrm{vol}(G)/2$. The sequence of visited nodes by this process is the dense branch of $\mathcal{T}$. Formally, for any HC tree $\mathcal{T}$ of $G$, the dense branch is the path $(A_0, A_1, \ldots, A_k)$ in $\mathcal{T}$ for some $k \in \mathbb{Z}_{\geq 0}$, such that the following conditions hold: (1) $A_0$ is the root of $\mathcal{T}$; (2) $A_k$ is the node such that $\mathrm{vol}(A_k) > \mathrm{vol}(G)/2$ and both children of $A_k$ have volume at most $\mathrm{vol}(G)/2$. It is important to note that the dense branch of a tree $\mathcal{T}$ is unique and it contains all nodes $A_i$ such that $\mathrm{vol}(A_i) > \mathrm{vol}(G)/2$ and only those nodes.

At a very high level, the proof of Theorem 1 starts with any optimal HC tree $\mathcal{T}_0$ of $G$ and constructs trees $\mathcal{T}_1, \mathcal{T}_2, \mathcal{T}_3$ and $\mathcal{T}_4$ such that (i) $\mathrm{cost}_G(\mathcal{T}_i)$ can be upper bounded with respect to $\mathrm{cost}_G(\mathcal{T}_{i-1})$ for every $1 \leq i \leq 4$, (ii) the final constructed $\mathcal{T}_4$ is exactly the tree $\mathcal{T}_{\mathrm{deg}}(G)$, the output of Algorithm 1. Combining these two facts allows us to upper bound $\mathrm{cost}_G(\mathcal{T}_{\mathrm{deg}})$ with respect to $\mathsf{OPT}_G$.

*Step 1: Regularisation.* Let $(A_0, \ldots, A_{k_0})$ be the dense branch of any optimal $\mathcal{T}_0$, and let $B_i$ be the sibling of $A_i$, for all $1 \leq i \leq k_0$. Let $i_{\min} = \lfloor \log_2(|A_{k_0}|) \rfloor$ and $i_{\max} = \lfloor \log_2(|V| - 1) \rfloor$. For simplicity, we assume that the dense branch has size at least 2, that $|A_1| \geq 2^{i_{\max}}$ and that $|A_{k_0}| = 2^{i_{\min}}$, which ensures that $i_{\min} \leq i_{\max}$. These assumptions allow us to better present the proof techniques by ignoring some corner cases. In the full version of our paper we deal with the

---

[3]To see this, notice that $\mathrm{cost}(\mathcal{T}^*) \leq \mathrm{cost}(\mathcal{T}^*[S]) + n \cdot \mathrm{vol}(G_1) + n^2 = \Theta(n^2)$, as the complete subgraph $G[S]$ induces a cost of $\Theta((n^{2/3})^3)$ [12]. The subgraph $G[S]$ also implies that $\mathsf{OPT}_G = \Omega(n^2)$.

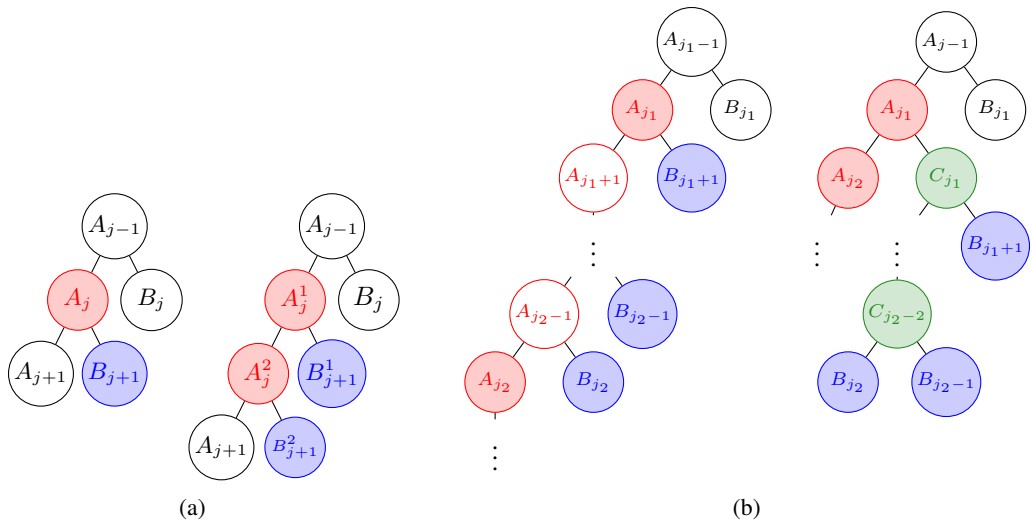

Figure 2: (a) With a proper partition of $B_{j+1}$, we have a new node $A_j^2$ of size $2^i$; (b) The nodes between $A_{j_1}$ and $A_{j_2}$ (left) have size in $(2^{i-1}, 2^i)$; These nodes are compressed (right) and only the nodes of size some power of 2 remain in the dense branch.

general case when some of these assumptions do not hold. The goal of this step is to adjust the dense branch of $\mathcal{T}_0$, such that the resulting tree $\mathcal{T}_1$ satisfies the following condition: for all $i \in [i_{\min}, i_{\max}]$, there is a node of size exactly $2^i$ along the dense branch of $\mathcal{T}_1$. We achieve this by applying a sequence of operations, each of which creating a new node of exact size $2^i$ for some suitable $i$. Concretely, let $i \in [i_{\min}, i_{\max}]$ be the largest integer such that there is no node of size $2^i$ on the dense branch of $\mathcal{T}_0$. As $|A_{k_0}| = 2^{i_{\min}}$, there is some node $A_j$ on the dense branch such that $|A_j| > 2^i$ and $|A_{j+1}| < 2^i$. We adjust the branch at $A_j$ as follows: (i) we consider a partition of $B_{j+1} = B_{j+1}^1 \cup B_{j+1}^2$ such that $|A_{j+1}| + |B_{j+1}^2| = 2^i$; (ii) we replace the node $A_j$ by some newly created node $A_j^1$ that has children $B_{j+1}^1$ and a new node $A_j^2$. (iii) the two children of $A_j^2$ will be $A_{j+1}$ and $B_{j+1}^2$. This adjustment is illustrated in Figure 2(a), and we repeat this process until no such $i$ exists any more. We call the resulting tree $\mathcal{T}_1$, and the following lemma gives an upper bound of $\text{cost}_G(\mathcal{T}_1)$.

**Lemma 3.2.** *Our constructed tree $\mathcal{T}_1$ satisfies* $\text{cost}_G(\mathcal{T}_1) \leq (1 + 4/\Phi_G) \cdot \text{cost}_G(\mathcal{T}_0)$.

*Step 2: Compression.* With potential relabelling of the nodes, let $(A_0, \ldots, A_{k_1})$ be the dense branch of $\mathcal{T}_1$ for some $k_1 \in \mathbb{Z}_+$ satisfying $|A_{k_1}| = 2^{i_{\min}}$, and $B_i$ be the sibling of $A_i$. The objective of this step is to ensure that, by a sequence of adjustments, all the nodes along the dense branch are of size equal to some power of 2. Here, we describe how one such adjustment is performed, and refer the reader to Figure 2(b) for illustration. Let $i \in [i_{\min}, i_{\max}]$ be some index such that $|A_{j_1}| = 2^i$ and $|A_{j_2}| = 2^{i-1}$ for some $j_1 < j_2$. We compress the dense branch by removing all nodes between $A_{j_1}$ and $A_{j_2}$ as follows: The two children of $A_{j_1}$ will be $A_{j_2}$ and some new node $C_{j_1}$, which has children $C_{j_1+1}$ and $B_{j_1+1}$; The two children of $C_{j_1+1}$ will be some new node $C_{j_1+2}$ and $B_{j_1+2}$, etc. The last node $C_{j_2-2}$ has children $B_{j_2-1}$ and $B_{j_2}$. In addition, we perform one more such adjustment to remove all nodes $A_j$ of size $2^{i_{\max}} < |A_j| < n$ and ensure all nodes (except potentially $A_0$) on the dense branch have size $2^i$ for some $i \in [i_{\min}, i_{\max}]$. We call the resulting tree $\mathcal{T}_2$, and the following lemma bounds the cost of $\mathcal{T}_2$.

**Lemma 3.3.** *Our constructed tree $\mathcal{T}_2$ satisfies that* $\text{cost}_G(\mathcal{T}_2) \leq 2 \cdot \text{cost}_G(\mathcal{T}_1)$.

*Step 3: Matching.* Let $(A_0, \ldots, A_{k_2})$ be the dense branch of $\mathcal{T}_2$, for some $k_2 \in \mathbb{Z}_+$, and $B_i$ be the sibling of $A_i$. In this step we transform $\mathcal{T}_2$ into $\mathcal{T}_3$, such that $\mathcal{T}_3$ is isomorphic to $\mathcal{T}_{\deg}(G)$, which ensures that $\mathcal{T}_3$ and $\mathcal{T}_{\deg}(G)$ have the same structure. To achieve this, for every $1 \leq i \leq k_2$ we simply replace each $\mathcal{T}_2[B_i]$ with $\mathcal{T}_{\deg}(G\{B_i\})$. We further replace $\mathcal{T}_2[A_{k_2}]$ with $\mathcal{T}_{\deg}(G\{A_{k_2}\})$. We call the resulting tree $\mathcal{T}_3$, and bound its cost by the following lemma:

**Lemma 3.4.** *Our constructed tree $\mathcal{T}_3$ satisfies that* $\text{cost}_G(\mathcal{T}_3) \leq (1 + 4/\Phi_G) \text{cost}_G(\mathcal{T}_2)$.

*Step 4: Sorting.* We assume that $(A_0, \ldots, A_{k_3})$ is the dense branch of $\mathcal{T}_3$ for some $k_3 \in \mathbb{Z}_+$, and we extend the dense branch to $(A_0, \ldots, A_{i_{\max}})$ with the same property that for every $i \in [k_3, i_{\max}]$ $A_i$ is the child of $A_{i-1}$ with the higher volume, and let $\mathcal{S} \triangleq \{B_1, \ldots, B_{i_{\max}}, A_{i_{\max}}\}$. Recall that in $\mathcal{T}_{\deg}(G)$ the first $r = 2^{i_{\max}}$ vertices of the highest degrees, i.e., $\{v_1, \ldots, v_r\}$ belong to $A_1$, of which the first $2^{i_{\max}-1}$ vertices belong to $A_2$, and so on; however, this might not be the case for $\mathcal{T}_3$. Hence, in the final step, we prove that $\mathcal{T}_3$ can be transformed into $\mathcal{T}_{\deg}(G)$ without a significant increase of the total cost. In this step we will swap vertices between the internal nodes $B_1, B_2, \ldots, B_{i_{\max}}, A_{i_{\max}}$ in such a way that $A_{i_{\max}}$ will consist of $v_1$ and $v_2$, $B_{i_{\max}}$ will consist of $v_3$ and $v_4$, $B_{i_{\max}-1}$ will consist of $v_5$ up to $v_8$, etc. We call vertex $u$ *misplaced* if the position of $u$ is $\mathcal{T}_3$ is different from the one in $\mathcal{T}_{\deg}(G)$. To transform $\mathcal{T}_3$ into $\mathcal{T}_{\deg}(G)$ we perform a sequence of operations, each of which consisting in a chain of swaps focusing on the vertex of the *highest* degree that is currently misplaced. For the sake of argument, we assume that $v_1 \notin A_{i_{\max}}$ is misplaced, and we apply the following operation to move $v_1$ to $A_{i_{\max}}$: (i) let $v_1 \in B_{i_0}$ for some $i_0 \geq 1$, and let $y$ be the vertex of the lowest degree among the vertices in $\mathcal{S} \setminus \{B_1, \ldots, B_{i_0}\}$. Say $y \in B_{i_1}$ for some $i_1 > i_0$, and swap $v_1$ with $y$ while keeping the structure of the tree unchanged; (ii) repeat the swap operation above until $v_1$ reaches its correct place $A_{i_{\max}}$. Once the above process is complete and $v_1$ reaches $A_{i_{\max}}$, we apply a similar chain of swaps to $v_2$ to ensure $v_2$ also reaches $A_{i_{\max}}$. Then, we sequentially apply the process to $v_3$ and $v_4$ to ensure they reach $B_{i_{\max}}$, and continue this process until there are no more misplaced vertices.

We call the resulting tree $\mathcal{T}_3'$, and notice that every node $X \in \mathcal{S}$ in $\mathcal{T}_3'$ contains the correct set of vertices. However, the positions of these vertices in $\mathcal{T}_3'[X]$ might be different from the ones in $\mathcal{T}_{\deg}(G\{X\})$. To overcome this issue, we repeat Step 3 again to the tree $\mathcal{T}_3'$, and this will introduce another factor of $(1 + 4/\Phi_G)$ to the total cost of the constructed tree. Importantly, the final constructed tree after this step is exactly the tree $\mathcal{T}_{\deg}(G)$ and we bound its cost by the following lemma:

**Lemma 3.5.** *It holds for $\mathcal{T}_{\deg}$ that* $\mathrm{cost}(\mathcal{T}_{\deg}) \leq (1 + 24/\Phi_G)(1 + 4/\Phi_G) \cdot \mathrm{cost}(\mathcal{T}_3)$.

Finally, by combining Lemmas 3.2, 3.3, 3.4 and 3.5 we prove the approximation guarantee of $\mathcal{T}_{\deg}$ in Theorem 1. The runtime of Algorithm 1 follows by a simple application of the master theorem.

## 4 Hierarchical clustering for well-clustered graphs

So far we have shown that an $O(1)$-approximate tree can be easily constructed for expander graphs. We will now focus on a wider class of well-clustered graphs. Informally, a well-clustered graph is a collection of densely-connected components (clusters) of high conductance, which are weakly interconnected. As these graphs form some of the most meaningful objects for clustering in practice, one would naturally ask whether our $O(1)$-approximation for expanders can be extended to well-clustered graphs. In this section, we will give an affirmative answer to this question.

To formalise the well-clustered property, we consider the notion of $(\Phi_{\mathrm{in}}, \Phi_{\mathrm{out}})$-*decomposition* introduced by Gharan and Trevisan [13]. Formally, for a graph $G = (V, E, w)$ and $k \in \mathbb{Z}_+$, we say that $G$ has $k$ well-defined clusters if $V(G)$ can be partitioned into disjoint subsets $\{P_i\}_{i=1}^k$, such that the following hold: (i) there's a sparse cut between $P_i$ and $V \setminus P_i$ for any $1 \leq i \leq k$, which is formulated as $\Phi_G(P_i) \leq \Phi_{\mathrm{out}}$; and (ii) each induced subgraph $G[P_i]$ has high conductance $\Phi_{G[P_i]} \geq \Phi_{\mathrm{in}}$. We underline that, through the celebrated higher-order Cheeger inequality [17], this condition of $(\Phi_{\mathrm{in}}, \Phi_{\mathrm{out}})$-decomposition can be approximately reduced to other formulations of a well-clustered graph studied in the literature, e.g., [23, 26, 27].

The starting point of our second result is the following polynomial-time algorithm presented by Gharan and Trevisan [13], which produces a $(\Phi_{\mathrm{in}}, \Phi_{\mathrm{out}})$-decomposition of graph $G$ for some parameters $\Phi_{\mathrm{in}}$ and $\Phi_{\mathrm{out}}$. Specifically, given a well-clustered graph as input, their algorithm returns disjoint sets of vertices $\{P_i\}_{i=1}^{\ell}$ with bounded $\Phi(P_i)$ and $\Phi_{G[P_i]}$ for each $P_i$, and the algorithm's performance is as follows:

**Lemma 4.1** ([13], Theorem 1.6). *Let $G = (V, E, w)$ be a graph such that $\lambda_{k+1} > 0$, for some $k \geq 1$. Then, there is a polynomial-time algorithm that finds an $\ell$-partition $\{P_i\}_{i=1}^{\ell}$ of $V$, for some $\ell \leq k$, such that the following hold for every $1 \leq i \leq \ell$:*

(A1) $\Phi(P_i) = O\left(k^6 \sqrt{\lambda_k}\right)$;

$(A2)$ $\Phi_{G[P_i]} = \Omega\left(\lambda_{k+1}^2/k^4\right).$

Informally, this result states that, when the underlying input graph $G$ presents a clear structure of clusters, one can find in polynomial-time a partition $\{P_i\}_{i=1}^{\ell}$ such that both the outer and inner conductance of every $P_i$ can be bounded. One natural question raising from this partition $\{P_i\}_{i=1}^{\ell}$ is whether we can directly use $\{P_i\}_{i=1}^{\ell}$ to construct an HC tree. As an obvious approach, one could consider to (i) construct trees $\mathcal{T}_i = \mathcal{T}_{\deg}(G[P_i])$ for every $1 \le i \le \ell$, and (ii) merge the tres $\{\mathcal{T}_i\}$ in the best way to construct the final tree $\mathcal{T}_G$. Unfortunately, this direct approach does not work, and we present in the full version of the paper an example where this approach fails to achieve an $O(1)$-approximation[4]. To address this, we follow our intuition gained from Figure 1, and further decompose every $P_i$ into smaller subsets. Similar with analysing *dense branches*, we introduce the *critical nodes* associated with each $\mathcal{T}_i$.

**Definition 4.2** (Critical nodes). *Let $\mathcal{T}_i = \mathcal{T}_{\deg}(G[P_i])$ be the tree computed by Algorithm 1 to the induced graph $G[P_i]$. Suppose $(A_0, \ldots, A_{r_i})$ is the dense branch of $\mathcal{T}_i$, for some $r_i \in \mathbb{Z}_+$, $B_j$ is the sibling of $A_j$ and let $A_{r_i+1}, B_{r_i+1}$ be the two children of $A_{r_i}$. We define $\mathcal{S}_i \triangleq \{B_1, \ldots, B_{r_i+1}, A_{r_i+1}\}$ and call every node $N \in \mathcal{S}_i$ a critical node.*

We remark that each critical node $N \in \mathcal{S}_i$ ($1 \le i \le \ell$) is an internal node of maximum size in $\mathcal{T}_i$ that is not on the dense branch. Moreover, each $\mathcal{S}_i$ is a partition of $P_i$. Based on critical nodes, we present an improved decomposition algorithm, which is similar to the one in Lemma 4.1, and prove that the output quality of our algorithm can be significantly strengthened for hierarchical clustering. Specifically, in addition to satisfying $(A1)$ and $(A2)$, we prove that the total weight between each critical node $N \in \mathcal{S}_i$ and all the other clusters $P_j$, for all $i \ne j$, can be upper bounded. We highlight that this is one of the key properties that allows us to obtain our main result, and also suggests that the original decomposition algorithm in [13] might not suffice for our purpose.

**Lemma 4.3** (Strong Decomposition Lemma). *Let $G = (V, E, w)$ be a graph such that $\lambda_{k+1} > 0$ and $\lambda_k = O(k^{-12})$. Then, there is a polynomial-time algorithm that finds an $\ell$-partition of $V$ into sets $\{P_i\}_{i=1}^{\ell}$, for some $\ell \le k$, such that properties $(A1)$ and $(A2)$ hold for every $1 \le i \le \ell$. Moreover, for every cluster $P_i$ and every critical node $N \in \mathcal{S}_i$, it holds that*

$(A3)$ $w(N, V \setminus P_i) \le 6(k+1) \cdot \text{vol}_{G[P_i]}(N).$

To underline the importance of $(A3)$, recall that, in general, each subtree $\mathcal{T}_i$ cannot be directly used to construct an $O(1)$-approximate HC tree of $G$ because of the potential high cost of the crossing edges $E(P_i, V \setminus P_i)$. However, if the internal cost of $\mathcal{T}_i$ is high enough to compensate for the cost introduced for the crossing edges $E(P_i, V \setminus P_i)$, then one can safely use this $\mathcal{T}_i$ as a building block. This is one of the most crucial insights that leads us to design our final algorithm `PruneMerge`.

Now we are ready to describe the algorithm `PruneMerge`, and we refer the reader to Algorithm 2 for the formal presentation. At a high-level, our algorithm consists of three phases: `Partition`, `Prune` and `Merge`. In the `Partition` phase (Lines 1–2), the algorithm invokes Lemma 4.3 to partition $V(G)$ into sets $\{P_i\}_{i=1}^{\ell}$, and applies Algorithm 1 to obtain the corresponding trees $\{\mathcal{T}_i\}_{i=1}^{\ell}$. The `Prune` phase (Lines 4–11) consists of a repeated pruning process: for every such tree $\mathcal{T}_i$, the algorithm checks in Line 7 if the maximal possible cost of the edges $E(P_i, V \setminus P_i)$ (i.e., the LHS of the inequality in the `if`-condition) can be bounded by the internal cost of the critical nodes $N \in S_i$, up to a factor of $O(k)$.

- If so, the algorithm uses $\mathcal{T}_i$ as a building block and adds it to a global set of trees $\mathbb{T}$;
- Otherwise, the algorithm prunes the subtree $\mathcal{T}_i[N]$, where $N \in \mathcal{S}_i$ is the critical node closest to the root in $\mathcal{T}_i$, and adds $\mathcal{T}_i[N]$ to $\mathbb{T}$ (Line 11).

The process is repeated with the pruned $\mathcal{T}_i$ until either the condition in Line 7 is satisfied, or $\mathcal{T}_i$ is completely pruned. Finally, in the `Merge` phase (Lines 13–15) the algorithm combines the trees in $\mathbb{T}$ in a "caterpillar style" according to an increasing order of their sizes. The performance of this algorithm is summarised in Theorem 2.

**Theorem 2.** *Let $G = (V, E, w)$ be graph, and $k > 1$ such that $\lambda_{k+1} > 0$ and $\lambda_k = O(k^{-12})$. The algorithm `PruneMerge` runs in polynomial-time and constructs an HC tree $\mathcal{T}_{\text{PM}}$ of $G$ satisfying $\text{cost}_G(\mathcal{T}_{\text{PM}}) = O\left(k^{22}/\lambda_{k+1}^{10}\right) \cdot \text{OPT}_G$. In particular, when $\lambda_{k+1} = \Omega(1)$ and $k = O(1)$, the algorithm's constructed tree $\mathcal{T}_{\text{PM}}$ satisfies that $\text{cost}_G(\mathcal{T}_{\text{PM}}) = O(1) \cdot \text{OPT}_G$.*

---

[4]The example consists of two copies of the graph in Figure 1(a), connected by a sparse cut.

---

**Algorithm 2:** `PruneMerge`$(G, k)$

---

**Input:** A graph $G = (V, E, w)$, a parameter $k \in \mathbb{Z}_+$ such that $\lambda_{k+1} > 0$;
**Output:** An HC tree $\mathcal{T}_{\mathrm{PM}}$ of $G$;

**1** Apply the partitioning algorithm (Lemma 4.3) on input $(G, k)$ to obtain $\{P_i\}_{i=1}^{\ell}$ for some $\ell \leq k$;
**2** Let $\mathcal{T}_i = $ `HCwithDegrees`$(G[P_i])$;
**3** Initialise $\mathbb{T} = \emptyset$;
**4 for** *All clusters $P_i$* **do**
**5**     Let $\mathcal{S}_i$ be the set of critical nodes of $\mathcal{T}_i$;
**6**     **while** *$\mathcal{S}_i$ is nonempty* **do**
**7**         **if** $n \cdot \sum_{N \in \mathcal{S}_i} w(N, V \setminus P_i) \leq 6(k+1) \cdot \sum_{N \in \mathcal{S}_i} |\mathrm{parent}_{\mathcal{T}_i}(N)| \cdot \mathrm{vol}_{G[P_i]}(N)$ **then**
**8**             Update $\mathbb{T} \leftarrow \mathbb{T} \cup \mathcal{T}_i$ and $\mathcal{S}_i = \emptyset$;
**9**         **else**
**10**             Let $N, M$ be the two children of the root of $\mathcal{T}_i$ such that $N \in \mathcal{S}_i$;
**11**             Update $\mathbb{T} \leftarrow \mathbb{T} \cup \mathcal{T}_i[N]$, $\mathcal{S}_i \leftarrow \mathcal{S}_i \setminus \{N\}$ and $\mathcal{T}_i \leftarrow \mathcal{T}_i[M]$;

**12** Let $t = |\mathbb{T}|$ and $\mathbb{T} = \{\widetilde{\mathcal{T}}_1, \ldots, \widetilde{\mathcal{T}}_t\}$ be such that $|\widetilde{\mathcal{T}}_i| \leq |\widetilde{\mathcal{T}}_{i+1}|$, for all $1 \leq i < t$;
**13** Initialise $\mathcal{T}_{\mathrm{PM}} = \widetilde{\mathcal{T}}_1$;
**14 for** $i = 2 \ldots t$ **do**
**15**     Let $\mathcal{T}_{\mathrm{PM}}$ be the tree with $\mathcal{T}_{\mathrm{PM}}$ and $\widetilde{\mathcal{T}}_i$ as its two children;
**16 return** $\mathcal{T}_{\mathrm{PM}}$.

---

We remark that, although Algorithm 2 requires a parameter $k$ as input, we can apply the standard technique of running Algorithm 2 for different values of $k$ and returning the tree with the lowest cost. By introducing a factor of $O(k)$ to the algorithm's runtime, this ensures that one of the constructed trees by Algorithm 2 would always satisfy our promised approximation ratio.

## 5 Experiments

We experimentally evaluate the performance of our proposed algorithm, and compare it against the three well-known linkage heuristics for computing hierarchical clustering trees, and different variants of the algorithm proposed in [10], i.e. `Linkage++`, on both synthetic and real-world data sets. At a high level, `Linkage++` consists of the following three steps:

  (i) Project the input data points into a lower dimensional Euclidean subspace;

  (ii) Run the `Single Linkage` algorithm [11] until $k$ clusters are left;

  (iii) Run a `Density` based linkage algorithm on the $k$ clusters until one cluster is left.

Specifically, our algorithm `PruneMerge` will be compared against the following 6 algorithms:

  - `Average Linkage`, `Complete Linkage`, and `Single Linkage`: the three well-known linkage algorithms studied in the literature. We refer the reader to [11] for a complete description.

  - `Linkage++`, `PCA+` and `Density`: the algorithm proposed in [10], together with two variants also studied in [10]. The algorithm `PCA+` corresponds to running Steps (i) and (ii) of `Linkage++` until one cluster is left (as opposed to $k$ clusters), while `Density` corresponds to running Steps (i) and (iii) of `Linkage++`.

All algorithms were implemented in Python 3.8 and the experiments were performed using an Intel(R) Core(TM) i5-6500 CPU @ 3.20GHz processor, with 16 GB RAM. All of the reported costs below are averaged over 5 independent runs.

**Synthetic data sets.** We first compare the performance of our algorithm with the aforementioned other algorithms on synthetic data sets.

*Clusters of the same size.* Our first set of experiments employ input graphs generated according to random stochastic models, where all clusters have the same size. For our first experiment, we look at graphs generated from the standard Stochastic Block Model (SBM). We first set the number of clusters as $k = 3$, and the number of vertices in each cluster $\{P_i\}_{i=1}^3$ as $1,000$. We assume that any pair of vertices within each cluster is connected by an edge with probability $p$, and any pair of vertices from different clusters is connected by an edge with probability $q$. We fix the value $q = 0.002$, and consider different values of $p \in [0.04, 0.2]$. Our experimental results are illustrated in Figure 3(a).

For our second experiment, we consider graphs generated according to a hierarchical stochastic block model (HSBM) [11]. This model assumes the existence of a ground-truth hierarchical structure of the clusters. For the specific choice of parameters, we set the number of clusters as $k = 5$, and the number of vertices in each cluster $\{P_i\}_{i=1}^5$ as $600$. For every pair of vertices $(u, v) \in P_i \times P_j$, we assume that $u$ and $v$ are connected by an edge with probability $p$ if $i = j$; otherwise $u$ and $v$ are connected by an edge with probability $q_{i,j}$ defined as follows: (i) for all $i \in \{1, 2, 3\}$ and $j \in \{4, 5\}$, $q_{i,j} = q_{j,i} = q_{\min}$; (ii) for $i \in \{1, 2\}$, $q_{i,3} = q_{3,i} = 2 \cdot q_{\min}$; (iii) $q_{4,5} = q_{5,4} = 2 \cdot q_{\min}$; (iv) $q_{1,2} = q_{2,1} = 3 \cdot q_{\min}$. We fix the value $q_{\min} = 0.0005$ and consider different values of $p \in [0.04, 0.2]$ as illustrated in Figure 3(b). We remark that this choice of parameters resembles similarities with [10] and this is to ensure that the underlying clusters exhibit a ground truth hierarchical structure.

As reported in Figure 3, our experimental results for both sets of graphs are similar, and the performance of our algorithm is marginally better than `Linkage++`. This is well expected, as `Linkage++` is specifically designed for the HSBM, in which all the clusters have the same inner density characterised by parameter $p$, and their algorithm achieves a $(1 + o(1))$-approximation for those instances.

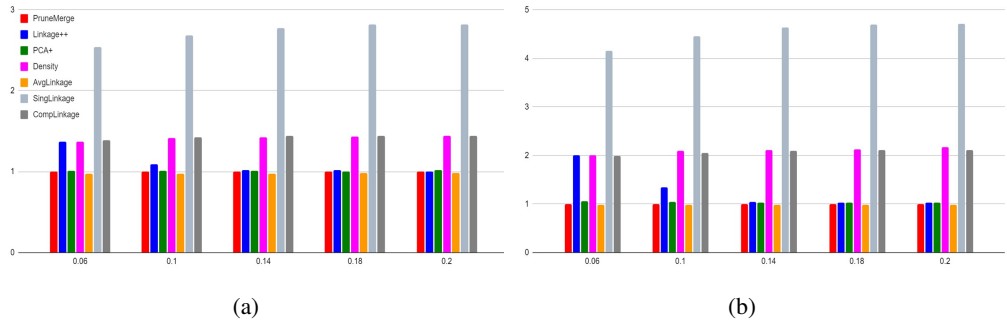

(a)             (b)

Figure 3: Results for clusters of the same size. The $x$-axis represents different values of $p$, while the $y$-axis represents the cost of the algorithms' returned HC trees normalised by the cost of `PruneMerge`. Figure (a) corresponds to inputs generated according to the SBM, while Figure (b) to those according to the HSBM.

*Clusters with non-uniform densities.* Next we study graphs in which edges are present *non-uniformly* within each cluster (e.g., Figure 1(a) discussed earlier). Specifically, we set $k = 3, |P_i| = 1000, q = 0.002, p = 0.06$, and every pair of vertices $(u, v) \in P_i \times P_j$ is connected by an edge with probability $p$ if $i = j$ and probability $q$ otherwise. Moreover, this time we choose a random set $S_i \subset P_i$ of size $|S_i| = c_p \cdot |P_i|$ from each cluster, and add edges to connect every pair of vertices in each $S_i$ so that the vertices of each $S_i$ form a clique. By setting different values of $c_p \in [0.05, 0.4]$, the performance of our algorithm is about $20\% - 50\%$ better than `Linkage++` with respect to the cost value of the constructed tree, see Figure 4(a) for detailed results. To explain the outperformance of our algorithm, notice that, by adding a clique into some cluster, the cluster structure is usually preserved with respect to $(\Phi_{\text{in}}, \Phi_{\text{out}})$ or similar eigen-gap assumption on well-clustered graphs. However, the existence of such a clique within some cluster would make the vertices' degrees highly unbalanced; as such many clustering algorithms that involve the matrix perturbation theory in their analysis might not work well.

*Clusters of different sizes.* To further highlight the significance of our algorithm on synthetic graphs of non-symmetric structures among the clusters, we study the graphs in which the clusters have different sizes. We choose the same set of $k$ and $q$ values as before ($k = 3, q = 0.002$), but set the sizes of the clusters to be $|P_1| = 1900, |P_2| = 900$ and $|P_3| = 200$. Every pair of vertices $u, v \in P_i$, for $i \in \{1, 2\}$ is connected by an edge with probability $p_1 = 0.06$, while pairs of vertices

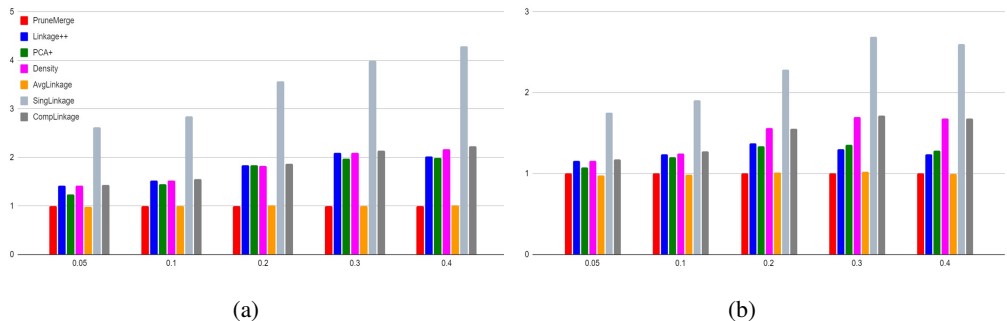

(a)               (b)

Figure 4: Results for graphs with non-uniform densities, or different sizes. The $x$-axis represents the $c_p$-values, while the $y$-axis represents the cost of the algorithms' constructed trees normalised by the cost of the ones constructed by `PruneMerge`. Figure (a) corresponds to inputs where all clusters have the same size, while in Figure (b) the clusters have different sizes.

$u, v \in P_3$ are connected with probability[5] $p_2 = 5 \cdot p_1 = 0.3$. We further plant a clique $S_i \subset P_i$ of size $|S_i| = c_p \cdot P_i$ for each cluster $P_i$, as in the previous set of experiments. By choosing different values of $c_p$ from $[0.05, 0.4]$, our results are reported in Figure 4(b), demonstrating that our algorithm performs better than the ones in [10].

**Real-world data sets.** To evaluate the performance of our algorithm on real-world data sets, we follow the sequence of recent work on hierarchical clustering [2, 10, 19, 24], all of which are based on the following 5 data sets from the Scikit-learn library [22] as well as the UCI ML repository [1]: Iris, Wine, Cancer, Boston and Newsgroup[6]. Similar with [24], for each data set we construct the similarity graph based on the Gaussian kernel, in which the $\sigma$-value is chosen according to the standard heuristic [21]. As reported in Figure 5, our algorithm performs marginally worse than `Linkage++` and significantly better than `PCA+`.

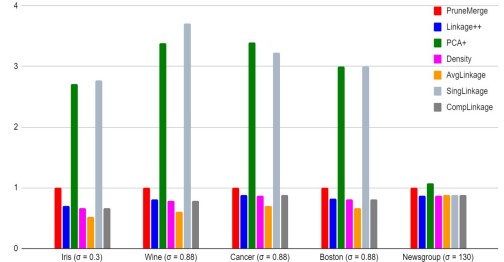

Figure 5: Results on real-world data sets. The $x$-axis represents the various data sets and our choice of the $\sigma$-value used for constructing the similarity graphs. The $y$-axis corresponds to the cost of the algorithms' output normalised by the cost of `PruneMerge`.

## 6 Conclusion

The experimental results on synthetic data sets demonstrate that our presented algorithm `PruneMerge` not only has excellent theoretical guarantees, but also produces output of lower cost than the previous algorithm `Linkage++`. In particular, the outperformance of our algorithm is best illustrated on graphs whose clusters have asymmetric internal structure and non-uniform densities. On the other side, the experimental results on real-world data sets show that the performance of `PruneMerge` is inferior to `Linkage++` and especially to `Average Linkage`. We believe that developing more efficient algorithms for well-clustered graphs is a very meaningful direction for future work.

Finally, our experimental results indicate that the `Average Linkage` algorithm performs extremely well on all instances, when compared to `PruneMerge` and `Linkage++`. This leads to the open question whether `Average Linkage` achieves an $O(1)$-approximation for well-clustered graphs, although it fails to achieve an $O(1)$-approximation for general graphs [11]. In our point of view, the answer to this question could help us design more efficient algorithms for hierarchical clustering that not only work in practice, but also have rigorous theoretical guarantees.

---

[5]Such choice of $p_2$ is to compensate for the small size of cluster $P_3$, and this ensures that the outer conductance $\Phi_G(C_3)$ is low.

[6]Due to the very large size of this data set, we consider only a subset consisting of "comp.graphics", "comp.os.ms-windows.misc", "comp.sys.ibm.pc.hardware", "comp.sys.mac.hardware", "rec.sport.baseball", and "rec.sport.hockey".

## Acknowledgements

Bogdan Manghiuc is supported by an EPSRC Doctoral Training Studentship (EP/R513209/1), and He Sun is supported by an EPSRC Early Career Fellowship (EP/T00729X/1).

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
