# OpenReview forum: "Hierarchical Clustering: $O(1)$-Approximation for Well-Clustered Graphs"
_NeurIPS.cc/2021/Conference — NeurIPS 2021 Poster_

### Official Review · Reviewer_Wup3 · 2021-07-07

**Rating:** 7
**Confidence:** 4

**Summary:**

The paper studies the problem of finding a good hierarchical clustering with respect to Dasgupta's objective function.

In particular, it gives algorithms that achieve constant-factor approximation for:
a) graphs that have good expansion properties
b) well-clusterable graphs, that is graphs that can be naturally clustered into sets of good expansion

These results are in contrast to known hardness results which rule out constant factor approximation under rather natural assumptions. Also, the algorithm generalizes an earlier result, which assumed much stronger properties of the input graph.
The paper also evaluates the new algorithm empirically, comparing it to a few existing baselines.

**Main Review:**

## Strengths

* The paper  broadens our understanding of the Dasgupta objective, and for this reason I consider it an important contribution to the area of hierarchical clustering. For example, the algorithm presented in the paper for the 'expander' case only uses the node degrees.
* I find the theoretical contribution highly nontrivial. On one hand, the fact that good expansion implies that the problem is tractable is not surprising (at least to me). However, turning this into an algorithm that also works in the case of clusterable graphs takes some highly nontrivial technical work.
* The paper is well-written and generally clear.

## Weaknesses
* The empirical evaluation says the algorithm improves upon "state-of-the-art", but I think this may be misleading, as the algorithm from the paper is typically outperformed by average-linkage clustering. I think the paper should comment on this and try to analyze the phenomenon, instead of ignoring the issue entirely.
* The result is unsurprising, in the sense that it's expected that by assuming stronger conditions on the input, one can obtain a better approximation guarantees. Still, making this idea work looks like solid piece of work.

## Specific comments

* Line 27: Is some word missing between 'is' and 'with'?
* Line 68: O(1 + o(1)) can be simplified to O(1). Do you mean O(1) or 1+o(1)?
* line 153: I think that condition (2) is redundant, as it is implied by condition (3)
* Lemma 4.1: Is \ell a parameter or something that the algorithm computes?

## Questions for authors
* Please analyze the empirical results taking average linkage into account, and be open about what it shows and what it does not. The analyzed algorithm is theoretical in nature, and I highly doubt one can show this can be a competitive method in practice. One approach would be to focus on comparison to the previous theoretical algorithm, include hierarchical stochastic block model in the studied datasets, and comment that average/single linkage are only added for reference. Right now, the writing in the empirical section is highly misleading to me, i.e., suggests the proposed method *is* competitive in practice, which, in my understanding, is contradicted by the pictures.

**Time Spent Reviewing:**

2

---

> ### Author Response · Authors · 2021-08-06
> **Response to Reviewer Wup3**
>
> Many thanks for your time and very positive review of our work. We have addressed each of your specific points below. Please let us know if you have any further questions and we will be happy to discuss them.
>
> ==== On the use of “state-of-the-art” (Point 1 of Weakness) ===
>
> In the submission the previous state-of-the-art refers to Cohen-Addad et al. [11] since, prior to our work, [11] is the only algorithm specifically designed for graphs consisting of multiple clusters with theoretically proven approximation guarantee. We didn’t view the average-linkage (AL) as the state-of-the-art, because there is no theoretical approximation guarantee of AL with respect to Dasgupta’s cost function, even in the case of well-clustered graphs. It wasn’t our intention to ignore the issue entirely, and we will reformulate our discussion in the final version to make our point clearer.
>
> === On the specific comments ===
>
> Line 27: There is no missing word between “is” and “with”, the meaning of the sentence being that the approximation guarantee depends on the conductance of $G$. We will rewrite this sentence if it would improve the readability in your point of view.
>
> Line 68:  This is indeed a typo, and it should be $(1+o(1))$.
>
> Line 153: You are right that the node $A_k$ referred in condition (3) is unique, thus making the dense branch well defined even if condition (2) is removed. However, we feel that the current form of introducing the dense branch improves the understanding for the reader unfamiliar with this concept.  We will consider to drop condition (2) if you think that would make the overall presentation cleaner.
>
> Lemma 4.1: $\ell$ is the number of clusters returned by the partitioning algorithm, hence the parameter that the algorithm computes. It is not a parameter that out algorithm sets.
>
>
> === On the questions for authors ===
>
> In the final version, we will make it clearer that the average/single linkage are only added for reference, and the main contribution of our work is mainly the theoretical one, rather than presenting some competitive algorithm in practice. We will also include hierarchical stochastic block model in the studied databases.

---

> > ### Comment · Reviewer_Wup3 · 2021-08-09
> > **The comments have been addressed**
> >
> > Thank you for addressing the comments. My (good) opinion of the paper remains unchanged.
> >
> > Very minor points:
> >
> > * the comment about "is with" was an oversight on my side, I don't think it requires fixing.
> > * It may be useful to add a note that condition (2) is implied by other conditions (but included for readability) or turning it into a corollary

---

> > > ### Author Response · Authors · 2021-08-23
> > > **Thank you**
> > >
> > > Many thanks for reading our response, and we are glad to see that you have kept a good opinion of our work.
> > >
> > > In the final version, we will carefully address your comment on condition (2), by either turning it into a corollary or adding a footnote with more explanation. Thanks a lot for helping us to improve the formulation of this technical condition.

---

### Official Review · Reviewer_64iM · 2021-07-16

**Rating:** 7
**Confidence:** 3

**Summary:**

This paper studies the cost function introduced by Dasgupta [13] and present polynomial-time approximation algorithms for well-clustered graphs. The well-clusterable assumptions mostly include an underlying structure of a fixed number of clusters, such that the subgraph induced by each cluster has high conductance, yet the cut between the clusters are sparse. The algorithm design bypasses the popular sparsest-cut subroutine, which was first proposed in Dasgupta [13] and often used by the community when studying this particular objective. The result is also more general than the state-of-the-art constant approximation results for this objective. The proposed algorithm also turns out to have reasonable performance empirically.

**Limitations And Societal Impact:**

The authors have adequately addressed the limitations. The potential negative societal impact of their work was marked as N/A.

**Main Review:**

Overall I think this is a solid theory paper. Roughly speaking, the technical part of the paper is structured as follows:
- It starts with a discussion on graphs with high conductance. Such a graph can be though of as one single cluster for which there might not be a reasonable cut. With a motivating example (Figure 1), the authors proposed to group vertices of similar degrees first to reduce the hierarchical clustering cost.
- On such a graph, the authors designs Algorithm 1 which constructs the hierarchical clustering tree based on degrees, and proves it is a constant approximation by moving vertices around in the HC tree and slightly modifying the HC tree step by step. Such movements of vertices are centered around a "dense branch" which is a path in the HC tree, where the internal nodes represent clusters whose sizes are bigger than their siblings.
- The paper then considers a well-clustered graph which is composed of a given number of such high-conductance subgraphs (clusters), where the inter-cluster edges are more sparse than intra-cluster edges.

Originality: The paper contains novel ideas for performing theoretical analysis on HC algorithms and in particular for Dasgupta's objective. Some of the techniques used in the paper, including focusing on the "dense branch" and reconstructing trees by slighly modifying some of the branches, were also adopted by previous work, but this paper carefully managed to put them to good use, and, has found a nice breakthrough for HC-related theory, in my opinion. Analysis for Dasgupta's objective function often comes hand in hand with finding sparsest-cut, and the methods proposed in this paper at least offered a different perspective.

The submission seems to be technically sound. The writing quality is good. The underlying algorithms are quite complicated but the authors were able to at least convey the main ideas. In particular, they have motived the assumptions and the algorithm design very well.

Significance: The paper's main contributions are theoretical. Assuming it is correct, it adds to the community's understanding of HC objective functions and may contain some new tools that can be used for studying Dasgupta's objective. Emprically, the algorithm is on par with other algorithms developed in previous work. It does not outperform the most widey used HC algorithms yet, but the results may still enlighten us.

Minor Comments:
- In Algorithm 1, by definition, for some |V| values r could be very close to |V|? For example if |V| = 2^m + 1, does it mean r = |V| - 1? It seems a bit unusual then, since the algorithm splits off one point. I wasn't able to check the details due to time limits, but maybe the authors can share some intuition about why this is fine?
- The intuition behind the algorithm is to group vertices of similar degrees first, which is also a bit non-intuitive, as I cannot really tell why it might give good HC trees at the high level. It might work in this context, but I wonder whether it is easily extendable.

**Time Spent Reviewing:**

8 hrs

---

> ### Author Response · Authors · 2021-08-09
> **Response to Reviewer 64iM**
>
>  Many thanks for your time and the very positive opinion expressed in the review. We are also thankful for generously highlighting the original aspects and significance of our work. We have addressed each raised comment below. Please let us know if you have any further question and we will be happy to discuss them.
>
>
> ==== The first minor comment ===
>
> The answer to the both questions are yes; e.g., when $|V| = 2^a + 1$ for some $a \geq 0$, we have that $r = |V| - 1$. However, in this case, this "unusual" split happens only at the top level. More generally, if $|V| = 2^a + b$, for $0 < b < 2^a$, at the root level the algorithm trims off the bottom $b$ vertices of the lowest degree which constitute $B$,  and then constructs the perfect binary tree $T_1$ on the vertices of $A$, which is merged with the tree $T_2$ formed on the vertices of $B$. Since $B$ contains the vertices of the lowest degree, in most cases the cost of the final tree $T_{\mathrm{deg}}$ is (heavily) dominated by the cost of the tree $T_1$, hence the trimming at the top does not significantly contribute to the overall cost.
>
> Moreover, we want to emphasise that the final tree $T_{\mathrm{deg}}$ exhibits several nice properties extensively used in our analysis, e.g., every internal node $A_i$ on the dense branch of $T_{\mathrm{deg}}$ has size $|A_i| = 2^j$, for some $j$. Finally, we acknowledge an alternative construction where vertices are partitioned into two roughly equal sets at every level (including the root) might be possible. However, we believe this would add several technical nuances in the analysis that makes the overall presentation and proof more complicated.
>
> ==== The second minor comment ===
>
> This is a very good question, and we believe that the construction based on grouping vertices according to their degrees cannot be extended to graphs with low conductance. To gain some intuition on why it works for high conductance graphs, consider the following scenario where $G$ is a graph of conductance $\phi$ and suppose, for simplicity, that $|V| = 2^a$ for some $a > 0$. Consider an arbitrary HC-tree with the split $(A, B)$ at the root level such that $\mathrm{vol}(A) \geq \mathrm{vol}(B)$. By the definition of volume and the fact that $G$ has conductance $\phi$, we know that $\phi \cdot \mathrm{vol}(B) \leq w(A, B) \leq \mathrm{vol}(B)$. Therefore, up to a factor of $\phi$, a small (or large) value of $\mathrm{vol}(B)$ implies a small (or large) cut value $w(A, B)$. In our construction, we choose $B$ as the set of the $|V|/2$ vertices of the lowest degree. This ensures that: (1) the induced cut $(A, B)$ is balanced and (2) the set $B$ has smallest volume among all subsets of $|V|/2$ vertices, implying a small cut value $(A, B)$. Therefore, sequentially grouping the vertices in this fashion ensures that the cost induced at every level of the tree is not too large.

---

> > ### Comment · Reviewer_64iM · 2021-08-23
> > **Acknowledgement**
> >
> > Thanks to the authors for responding. The rebuttal shows that the authors have put a lot of consideration into this work, and have good understanding about hierarchical clustering. I will keep my positive opinions of the paper.

---

> > > ### Author Response · Authors · 2021-08-24
> > > **Thank you**
> > >
> > > Many thanks for reading our response and appreciation of our work. We hope that we have answered your detailed questions, and we are glad to see that you will keep your positive opinions of our work.

---

### Official Review · Reviewer_TUMr · 2021-07-16

**Rating:** 7
**Confidence:** 4

**Summary:**

Dasgupta's is a popular objective for hierarchical clustering. Given graph of similarities (weights w_uv), find the smallest subtree T_uv containing both u and v. The objective is \sum w_uv |T_uv|.

This can be optimized to within \sqrt \log n approximation, and constant factor is thought to be hard to achieve in general. There are recent work showing that this goal is possible for special graphs.

Cohen-Addad et al. present 1+o(1) approximation for a hierarchical generalization of SBM. This work takes a different perspective and looks at well-clusterable graphs in terms of conductance: There exists a partition V_i of the vertices into k parts such that (1) the subgraph induced on each V_i has large ("inner") conductance, and (2) each V_i has low ("outer") conductance. They show that one can get approximations depending on K and the conductance bounds. In particular, this gives const factors for expanders in the case of small k.

First they have a simple lemma with an upper bound for cost of any hierarchy in terms of conductance and degree distribution (ratio of d_max over d_avg and ratio of d_avg over d_min). This already gives const factor for const-degree expanders. Planting a large clique makes the bound polynomial, though there is a simple solution with small cost. They present a very simple algorithm (basically by sorting according to degree and recursively splitting the tree almost equally) which gives const factor for expanders.

Next they move to well-clusterable graphs, and show that Gharan-Trevisan decomposition is not sufficient, as simply joining the clusters in a tree may not work to get a good value for the objective (due to high cut values). Instead, they augment the decomposition theorem to identify a set of "critical nodes" with potentially high cut values. Then they break some of the clusters (more precisely, the greedily built trees for clusters) at those points, and finally merge everything into one tree by means of their degree-sorting algorithm.

**Ethical Concerns:**

I don't think so

**Limitations And Societal Impact:**

Not really

**Main Review:**

Clustering well-clusterable graphs is a top topic, and this work certainly adds to the literature. The algorithms and theoretical bounds are not very practical, though. The gains in practice are small.

The paper is easy to read despite the heavy math.

**Time Spent Reviewing:**

2

---

> ### Author Response · Authors · 2021-08-06
> **Response to Reviewer TUMr**
>
> Many thanks for your time and very positive review of our work. We acknowledge that the algorithms and theoretical bounds are not very practical, and hence their gains in practice might be small. On the other side, as our work is the first one demonstrating that $O(1)$-approximate HC trees can be constructed in polynomial-time for well-clustered graphs, we expect that this would inspire some future research on developing more practical algorithms of hierarchical clustering for well-clustered graphs. In our point of view, this is a very meaningful research direction, and will potentially have many practical applications.

---

> > ### Comment · Reviewer_TUMr · 2021-08-16
> > **Acknowledgement**
> >
> > Thanks for responding. This doesn't change my judgement. I like the paper and recommend it for acceptance.

---

> > > ### Author Response · Authors · 2021-08-23
> > > **Thank you**
> > >
> > > Many thanks for reading our response. We are glad to see that you like our work and recommend it for acceptance.

---

### Official Review · Reviewer_sgob · 2021-07-17

**Rating:** 5
**Confidence:** 4

**Summary:**

The authors study the important problem of clustering graph in a hierarchical way optimizing and present novel algorithms for the special case of graphs with expender structure or well clusterable graphs, The algorithms have state of the art approximation for such cases and in practice are implementable. The experimental evidence that the algorithms are in line with other baselins in synthetic graphs, and on real data have better performance than some theoretical algorithm (but worse than simple baselines like average linkage).

**Limitations And Societal Impact:**

All in all the paper is technically interesting but the experimental evidence shows that it does not improve over simple baseline algorithms like average and complete linkage this may suggest that the perhaps the assumption of well-clusterability is not practically useful in this problem on real data.


**Main Review:**


The authors study the important problem of clustering graph in a hierarchical way optimizing an objective function introduced by Dasgupta [13] .
The authors study the problem under the assumption that the graph is well clusterable, in particular that it has high conductance (O(1)) this means that the graph is an expander and has no sparse cut separating two parts of the graph. Under this assumption they show a very fast near linear algorithm constant factor algorithm (while state of the art algorithm have logarithmic approximation and use complex poly time subroutines). Then they generalize this to well clustered graphs going beyond the stochastic block model to arbitrary graphs with this property.

Moreover the paper is complemented with an empirical analysis.

The first structural observation is that if the graph has constant conductance and constant ratio of min max degree any tree is a constant approx solution. Then they introduce an algorithm which simply sorts nodes by degree and builds a tree balancing the nodes in order. Surprisingly this algorithm is a O(1/poly(conductance)) approximate despite being oblivious to the structure of the graph.
This suggests to me that getting a constant approximate hierarchical clustering of such graphs (despite the theoretical interest( is not an practically interesting result and as such the algorithm is not practically useful (expect as a building block).
Then they adapt their algorithm to well clustered graphs using results from Trevisan et al. on finding sparse cuts. The algorithm is non trivial,

Then they provide an empirical evaluation of their algorithm against standard heuristic baselines for HC and some algorithms with theoretical guarantees for special cases of the HC problem.  First they compare on synthetic SBM data where the performance ties with state of the art. On other synthetic data with multiple densities they improve over some baselines but the performance is tied with average linkage. Finally on real data their performance is better than some state off the art method (for HC in some objective and setting) but worse than the baselines of Average and complete linkage.


Minor:
very interesting theoretical fact, -> perhaps just interesting.

**Time Spent Reviewing:**

1.5

---

> ### Author Response · Authors · 2021-08-06
> **Response to Reviewer sgob**
>
> Many thanks for your time and helpful review of our work. We have addressed each of your specific points below. Please let us know if you have any further questions and we will be happy to discuss them.
>
> ===== On the assumption of the well-clusterability =====
>
> We highlight that our assumption of the well-clusterability and similar assumptions have been widely applied to characterise graphs occurring in practice (see the reference [14,18,22,23,26,27] of the submission). This is mainly because clustering is a hard problem to solve in general; on the other side, most instances for clustering that occur in practice usually exhibit a clear structure of clusters, so it’s natural to study clustering problems under some well-clusterability assumption.
>
> For the specific problem of hierarchical clustering, while achieving a constant-factor approximation  is SSE-hard for general graphs, our work shows that $O(1)$-approximation can be achieved in polynomial-time once the input graph is well-clustered. This is also appreciated by Reviewer Wup3, which comments that "These results are in contrast to known hardness results which rule out constant factor approximation under rather natural assumptions".
>
> ===== Comparison with the baselines of average and complete linkage =====
>
> We acknowledge your view that the experimental evidence shows that the output of our algorithm doesn't improve over simple baseline algorithms like average and complete linkage. However, we highlight that the objective of our work is to present algorithms with *theoretical approximation guarantees*; we remark that there are no theoretical guarantees for the above-mentioned linkage algorithms under Dasgupta’s cost function.
>
> ===== Wording issue=====
>
> We will follow your suggestion, and drop "very" before "interesting theoretical fact".

---

> > ### Comment · Reviewer_sgob · 2021-08-20
> > **Acknowledgement**
> >
> > I have read the rebuttal I suggest the authors if the paper is accepted to highlight in the experimental section that the work is theoretical in nature and that simple baselines (despite the lack of guarantees) often outperforms their algorithm. Also I would suggest to not claim that it exceeds the state of the art in practice.

---

> > > ### Author Response · Authors · 2021-08-23
> > > **Thank you**
> > >
> > > Many thanks for reading our response, and we hope that we have addressed your concern on the well-clusterability condition as no further concern has raised.
> > >
> > > In the final version of the paper, we will follow your suggestion and make it clear that our work is theoretical in natural and doesn't outperform the simple average and complete linkage algorithms. We will also make it more precise when using the term "state-of-the-art".

---

### Decision · Program_Chairs · 2021-09-27

**Decision:**

Accept (Poster)

**Comment:**

The authors present a new hierarchical algorithm to cluster graphs. The main result showed in the paper is that the new algorithm returns a constant approximation for a well-clustered graph. The paper is a clear extension of previous work in NeurIPS and it presents interesting theoretical results. One shortcoming of the current write-up is that the proposed algorithm does not have great experimental performances. Overall, the paper is interesting and proposes some novel ideas and it will be a nice contribution to NeurIPS.